# A unified framework for herbivore-to-producer biomass ratio reveals the relative influence of four ecological factors

Takehiro Kazama[1,9], Jotaro Urabe [1✉], Masato Yamamichi[2,10], Kotaro Tokita[1], Xuwang Yin[3], Izumi Katano[4,5], Hideyuki Doi [6], Takehito Yoshida[7,2] & Nelson G. Hairston Jr [8]

The biomass ratio of herbivores to primary producers reflects the structure of a community. Four primary factors have been proposed to affect this ratio, including production rate, defense traits and nutrient contents of producers, and predation by carnivores. However, identifying the joint effects of these factors across natural communities has been elusive, in part because of the lack of a framework for examining their effects simultaneously. Here, we develop a framework based on Lotka–Volterra equations for examining the effects of these factors on the biomass ratio. We then utilize it to test if these factors simultaneously affect the biomass ratio of freshwater plankton communities. We found that all four factors contributed significantly to the biomass ratio, with carnivore abundance having the greatest effect, followed by producer stoichiometric nutrient content. Thus, the present framework should be useful for examining the multiple factors shaping various types of communities, both aquatic and terrestrial.

[1] Aquatic Ecology Laboratory, Graduate School of Life Sciences, Tohoku University, 6-3 Aza-Aoba, Aramaki, Aoba-ku, Sendai, Miyagi 980-8578, Japan. [2] Department of General Systems Studies, University of Tokyo, 3-8-1 Komaba, Meguro, Tokyo 153-8902, Japan. [3] Liaoning Provincial Key Laboratory for Hydrobiology, College of Fisheries and Life Science, Dalian Ocean University, 52 Heishijiao Street, Shahekou District, Dalian 116023, China. [4] Graduate School of Humanities and Sciences, Nara Women's University, Kitanoya-nishimachi, Nara 630-8506, Japan. [5] KYOUSEI Science Center for Life and Nature, Nara Women's University, Kitanoya-nishimachi, Nara 630-8506, Japan. [6] Graduate School of Simulation Studies, University of Hyogo, 7-1-28 Minatojima-minamimachi, Chuo-ku, Kobe 650-0047, Japan. [7] Research Institute for Humanity and Nature, 457-4 Motoyama, Kamigamo, Kita-ku, Kyoto 603-8047, Japan. [8] Department of Ecology and Evolutionary Biology, Cornell University, Ithaca, NY 14853, USA. [9] Present address: Lake Biwa Branch Office, Center for Regional Environmental Research, National Institute for Environmental Studies, 5-34 Yanagasaki, Otsu, Shiga 520-0022, Japan. [10] Present address: School of Biological Sciences, The University of Queensland, Brisbane, QLD, Australia. ✉email: urabe@tohoku.ac.jp

The biomass ratio of herbivores ($H$) to primary producers ($P$) in nature varies by four orders of magnitude ($10^{-4}\sim10^1$)[1,2]. Because it reflects the structure of a community and ecosystem properties, such as energy flow from producers to higher trophic levels and nutrient cycling[1–4], a large number of studies have empirically and theoretically examined the $H/P$ biomass ratio and have shown that factors related with either bottom–up or top–down control may play crucial roles. These factors are production rate[5–7], defense traits[8–11], and nutrient content of the producers[2,4], and predation rate by carnivores including food-chain length[12–14]. However, it has been difficult to quantify how these factors act together to affect the $H/P$ biomass ratio in natural communities, because few studies have considered a theoretical framework for examining these effects simultaneously.

Top–down control of trophic structure is determined by the feeding rate and the abundance of consumers at higher trophic levels[15,16], whereas bottom–up control is determined by the primary production rate regulated by nutrient supply[17,18] and light[19]. In addition, both for terrestrial and aquatic producers including vascular plants and algae, chemical and physical defense traits are well documented as factors limiting herbivory[9,20,21], indicating that edibility of primary producers is a crucial factor in determining the biomass ratio[9,11]. The nutrient content of producers is also viewed as a primary factor in regulating $H/P$ mass ratio[2,4,22,23]. Depending on the supply rates of light and nutrients, the contents of biologically important elements such as nitrogen and phosphorus relative to carbon vary widely among primary producers[4]. As herbivore growth strongly depends on the elemental content of primary producers[22,23], the stoichiometric mismatch in carbon to phosphorus or carbon to nitrogen ratios between primary producers and herbivores likely results in a decrease in the $H/P$ ratio[2,4,22]. However, no study has yet examined how such a stoichiometric mismatch combined with other factors jointly affect the $H/P$ biomass ratio.

In this study, we used classic Lotka–Volterra equations[24,25] to develop a framework that simultaneously assesses the effects of primary production rate, producer defense traits and producer nutrient content, and predation rate on the H/P ratio in natural communities. We then test the model using data from communities inhabiting freshwater experimental ponds. In the ponds, the bottom–up factor (light) was manipulated by shading, while the top–down factor (fish predation) was quantified regularly. We show that, for pond plankton communities, top–down control and the stoichiometry of primary producers played pivotal roles in determining the $H/P$ ratio, followed by defense traits and the rate of primary production.

## Results

**A unified framework model of the H/P ratio.** Based on the Lotka–Volterra equations[24,25], the biomass dynamics of primary producers ($P$) and herbivores ($H$) are described as follows:

$$dP/dt = g(P)P - xP - f(P)PH,$$
$$dH/dt = kf(P)PH - mH \quad (1)$$

where $g(P)$ is biomass-specific primary production ($gC\,gC^{-1}\,d^{-1}$) and may be a function of $P$ ($gC\,m^{-2}$) owing to density-dependent growth, $f(P)$ is per capita grazing rate of herbivores ($m^2\,gC^{-1}\,d^{-1}$) and also may be function of $P$ depending on the functional response, $x$ is biomass-specific loss rate of primary producers other than due to grazing loss ($gC\,gC^{-1}\,d^{-1}$), $k$ is the conversion efficiency of herbivores as a fraction of ingested food converted into herbivore biomass (dimensionless: 0~1), and $m$ is per capita mortality rate of herbivores owing to predation and other factors ($gC\,gC^{-1}\,d^{-1}$). A list of model variables is listed in Supplementary Table 1. If we assume both $g(P)$

and $f(P)$ are constants, Eq. (1) is basically the Lotka–Volterra model, whereas it is an expansion of the Rosenzweig–MacArthur model if we assume logistic growth for $g(P)$ and Michaelis–Menten (Holling type II) functional responses for $f(P)$[24]. At the equilibrium state, i.e., $dP/dt = 0$ and $dH/dt = 0$, the abundance of producers ($P^*$) and consumers ($H^*$) can be represented as:

$$H^*/P^* = \{[g(P^*) - x]/f(P^*)\}/\{m/[kf(P^*)]\} \quad (2)$$

Thus, the relationship between $H$ and $P$ is not affected by the types of the functional response in herbivores ($f(P)$). If we set $g(P)$ as the biomass-specific primary production rate at equilibrium, as in simple Lotka–Volterra equations (i.e., $g(P^*) = g$), then the $H/P$ ratio can be expressed with log transformation as:

$$\log(H^*/P^*) = \log(k) + \log(g - x) - \log(m) \quad (3)$$

At equilibrium, $(g - x)P^*$ is the amount of primary production that herbivores consume per unit of time ($f(P^*)P^*H^*$). Thus, if this amount is divided by primary production per unit of time ($P^*g$), it corresponds to the fraction of primary production that herbivores consume (0~1). We define it as $\beta$ ($= 1 - x/g$). Large $\beta$ values imply that producers are efficiently grazed at the equilibrium state. Thus, $\beta$ is a gauge of inefficiency in the producers' defensive traits. Using these parameters, the $H^*/P^*$ ratio can be expressed as:

$$\log(H^*/P^*) = \log(k) + \log(\beta) + \log(g) - \log(m) \quad (4)$$

This equation implies that the $H^*/P^*$ biomass ratio on a log scale is affected additively by the specific primary production rate ($\log(g)$), the grazeable fraction of primary production ($\log(\beta)$), the conversion efficiency ($\log(k)$), and the mortality rate of herbivores ($\log(m)$). According to this equation, communities with relatively low carnivore abundance would have a correspondingly low value of $m$ and will exhibit high herbivore biomass relative to producer biomass ($H^*/P^*$), whereas those with low primary production (with low value of $g$) owing to, for example, low light supply will have a low $H^*/P^*$ ratio. An increase in defended producers such as armored plants or a decrease in edible producers will decrease $\beta$ by increasing the loss rate $x$ owing to the cost of defense, and will result in a decreased $H^*/P^*$ ratio. Finally, when the nutritional value of producers decreases, the conversion efficiency of herbivores ($k$) should be low, which in turn decreases the $H^*/P^*$ biomass ratio.

**The model for a test with plankton communities.** To apply Eq. (4) to a natural community, some modifications are necessary. Here, we consider a plankton community composed of algae and zooplankton. A theory of ecological stoichiometry suggests that the carbon content of primary producers relative to their nutrient content such as nitrogen or phosphorus is an important property affecting growth efficiency in herbivores[4]. Supporting the theory, a number of studies have shown that growth rate in terms of carbon accumulation relative to ingestion rate strongly depends on the carbon contents of the food relative to nutrients[26–29]. Thus, $k$ can be expressed as:

$$k = q_1 \times a_{nut}^{\varepsilon_1} \quad (5)$$

where $\alpha_{nut}$ is carbon content relative to nutrient content of primary producers and $q_1$ is the conversion factor adjusting to biomass units. In this study, we applied a power function with coefficient of $\varepsilon_1$ as a first order approximation because effects of this factor on the $H^*/P^*$ biomass ratio may not be proportionally related to plant nutrient content. For example, if $\varepsilon_1$ is much smaller than zero, it means that negative effects of the carbon to phosphorus ratio of algal food on an herbivore's $k$ are more substantial when the carbon to phosphorus ratio is high compared with the case when the carbon to phosphorus ratio is low.

However, if this factor does not affect the $H^*/P^*$ biomass ratio, $\varepsilon_1 = 0$ and $k$ is constant.

As herbivorous zooplankton cannot efficiently graze on larger phytoplankton due to gape limitation[30], the feeding efficiency of herbivores or the defense efficiency of the producers' resistance traits, $\beta$, would be related to the fraction of edible algae in terms of size as follows:

$$\beta = q_2 \times a_{edi}^{\varepsilon_2} \qquad (6)$$

where $\alpha_{edi}$ is a trait determining producer edibility, $q_2$ is a factor for converting the traits to edible efficiency, and $\varepsilon_2$ is how effective the trait is in defending against grazing. We expect $\varepsilon_2 = 0$ if this factor does not matter in regulating the $H^*/P^*$ biomass ratio but $\varepsilon_2 > 0$ if it has a role. Similarly, $g$ can be described as

$$g = q_3 \times \mu^{\varepsilon_3} \qquad (7)$$

where $\mu$ is the specific growth rate of producers, $q_3$ is a conversion factor, and $\varepsilon_3$ is the effect of $\mu$ on growth rate. Again, we expect that $\varepsilon_3 \neq 0$ if $g$ has a role in determining the $H^*/P^*$ ratio. Finally, assuming a Holling type I functional response of carnivores, the mortality rate of herbivores, $m$, is expressed as:

$$m = q_4 \times \theta^{\varepsilon_4} \qquad (8)$$

where $\theta$ is abundance of carnivores, $q_4$ is specific predation rate, and $\varepsilon_4$ is the effect size of carnivore abundance on $m$.

By inserting Eqs. (5–8) to Eq. (4), effects of factors on the $H^*/P^*$ biomass ratio is formulated as:

$$\log(H^*/P^*) = \varepsilon_1 \log(a_{nut}) + \varepsilon_2 \log(a_{edi}) + \varepsilon_3 \log(\mu) - \varepsilon_4 \log(\theta) + \gamma \qquad (9)$$

where $\gamma$ is $\log(q_1) + \log(q_2) + \log(q_3) - \log(q_4)$. If differences in the $H^*/P^*$ ratio among communities are regulated by growth rate ($\mu$), edibility ($\alpha_{edi}$), and nutrient contents ($\alpha_{nut}$) of producers as well as by predation by carnivores ($\theta$), we expect non-zero values for $\varepsilon_1 - \varepsilon_4$. Thus, Eq. (9) can be used to evaluate the relative importance of the four hypothesized agents if all of them simultaneously affect the $H^*/P^*$ ratio. Here we undertake this analysis using data for natural plankton communities in experimental ponds where primary production rate was manipulated with different abundance of carnivore fish.

**Experimental test by plankton communities.** The experiment was carried out at two ponds (pond ID 217 and 218) located at the Cornell University Experimental Ponds Facility in Ithaca, NY, USA during 4 June to 28 August 2016 (Fig. 1). Each pond has a 0.09 ha surface area (30 × 30 m) and is 1.5 m deep. To initiate the experiment, we equally divided each of the two ponds into four sections using vinyl-coated canvas curtains, and randomly assigned the four sections to either high-shade (64% shading), mid-shade (47% shading), low-shade (33% shading), or no-shade treatments (no shading). Shading in each treatment was made using opaque floating mats (6 m diameter; Solar-cell SunBlanket, Century Products, Inc., Georgia, USA)[31]. The floating mats were deployed silvered side up to reflect sunlight and blue side down to avoid pond heating. Sampling was performed biweekly for water chemistry and abundance of phytoplankton and zooplankton with measurements of vertical profiles of water temperature, dissolved oxygen (DO) concentration and photosynthetic active radiation (PAR).

PAR in the water column was lower in the sections with larger shaded areas throughout the experiment (Fig. 2b). Water temperature varied from 18 to 25 °C during the experiment but showed no notable differences in mean values of the water columns or the vertical profiles among the four treatments of the two ponds regardless of the shading treatment (Supplementary Fig. 1a, Supplementary Fig. 2). In all treatments, pH values gradually decreased towards the end of experiment and were higher in pond 218 (Supplementary Fig. 1b). DO concentration varied among the treatments and between the ponds, but were within the range of 5–12 mg L$^{-1}$ (Supplementary Fig. 1c).

Phytoplankton biomass (mg C L$^{-1}$) correlated significantly with chlorophyll $a$ (μg L$^{-1}$) ($r = 0.702$, $p < 0.001$), varied temporally (Supplementary Fig. 3), and was generally higher in the no-shade treatments, followed by the low-shade treatments in both Pond 217 and 218 (Fig. 2a). Zooplankton biomass also varied temporally (Supplementary Fig. 4) and was generally lower in low-shade treatments compared with other treatments (Fig. 2a). Although sampling began on 31 May, to remove effects of the initial conditions, we calculated mean phytoplankton ($P$) and zooplankton biomasses ($H$) in samples collected during the period from 10 June to 28 August 28 (Supplementary Table 2). Phytoplankton biomass was lower in Pond 218 regardless of the treatments, but such a notable difference between the ponds was not found in zooplankton biomass. Accordingly, no significant relationship was found between the mean values of phytoplankton and zooplankton biomass (Fig. 2a).

Both for zooplankton and phytoplankton, community composition was similar among the four treatments within the same pond (permutational multivariate analysis of variance (PERMANOVA), $F = 0.993$, $p = 0.42$ for algae; $F = 1.23$, $p = 0.34$ for zooplankton) but differed significantly between the two ponds ($F = 1.59$, $p = 0.017$ for algae; $F = 3.82$, $p = 0.047$ for zooplankton). In zooplankton communities, copepods dominated in pond 217, whereas large cladocerans including *Daphnia* occurred abundantly in pond 218 (Supplementary Fig. 5). In phytoplankton communities, Euglenophyceae and Chrysophyceae occurred abundantly in pond 217, and Euglenophyceae and Dinoflagellata

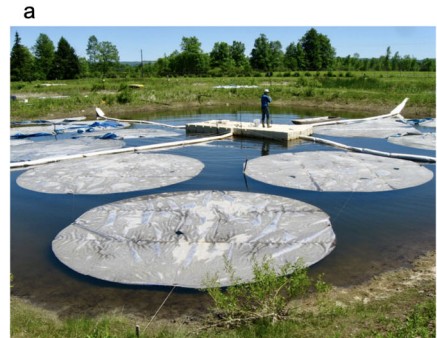
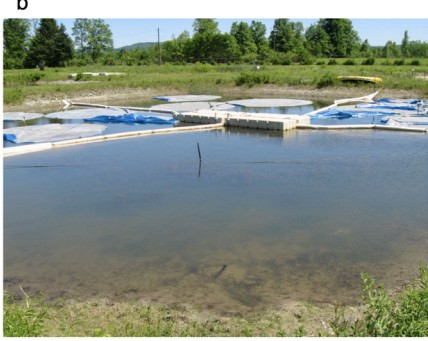

**Fig. 1 Ponds used in the experimental test.** Pond 217 (**a**) and Pond 218 (**b**) in the Cornell University Experimental Ponds Facility divided into four sections by vinyl-canvas curtains and partially shaded by floating mats to regulate primary production rate. Floating docks were placed at the center of the ponds for sampling.

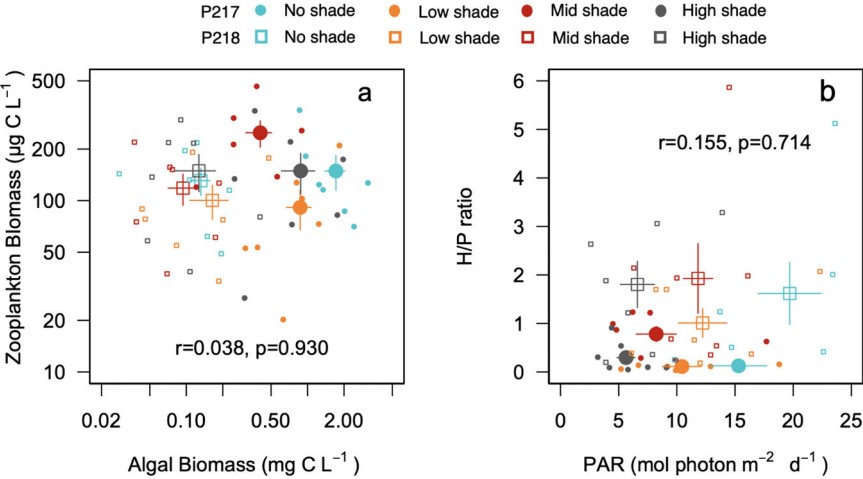

**Fig. 2 Relationships between zooplankton biomass and phytoplankton biomass, and between H/P mass ratio and photosynthetic active radiation.** Biplots of zooplankton biomass ($\mu$g C L$^{-1}$) and phytoplankton biomass (mg C L$^{-1}$) (**a**) and H/P mass ratio and photosynthetic active radiation (PAR, mol photon m$^{-2}$ d$^{-1}$) (**b**) in the water column during the experiment in no-shade (blue), low-shade (orange), mid-shade (red), and high-shade (gray) treatments in pond 217 (circles) and 218 (squares). In each panel, small symbols denote values at each sampling date, and large symbols denote the mean values among the sampling dates. Bars denote standard errors on the means ($n = 7$ sampling date in each section). Correlation coefficients ($r$) with $p$ values between the mean values are inserted in each panel.

dominated in pond 218 (Supplementary Fig. 6). In all treatments, cyanobacteria biomass was <20%. According to previous knowledge[30], we defined phytoplankton smaller than 30 $\mu$m (along the major axis of the cell or colony) as edible. Then, we calculated the fraction of the edible phytoplankton ($\alpha_{edi}$) as the ratio of edible phytoplankton biomass to total phytoplankton biomass, which varied from near zero to almost one in all the treatments of both ponds (Supplementary Fig. 3). The seston carbon to phosphorus ratio varied from 90 to 310 (Supplementary Fig. 7) and was higher for treatments with less shade in pond 217, whereas in pond 218 the seston carbon to phosphorus ratio did not vary among the treatments (Fig. 3b).

The chlorophyll *a* specific daily production rate estimated from the photosynthesis–PAR curve (Supplementary Fig. 8) varied temporally depending on weather conditions but was, in general, higher in treatments with less shade (Fig. 3(c)). Daily primary production rates also varied and were higher in treatments with less shade in pond 217, although in pond 218 the levels were similar among the treatments (Supplementary Fig. 4).

Fish abundance in each treatment section, determined as catch per unit of effort (CPUE) using minnow traps, showed that banded killifish (*Fundulus diaphanus*) and fathead minnow (*Pimephales promelas*) were present (Supplementary Fig. 9). Both fish species were collected on all sampling dates in pond 217 but were not caught after June 21 in pond 218 (Supplementary Fig. 4). Thus, mean abundance of these fish species was higher in pond 217 than in pond 218 (Fig. 3d). In the former pond, fish abundance also varied among the treatments, and was greater in no-shade treatments than in any of other treatments. Neither mean zooplankton biomass ($n = 8$, $r = 0.310$, $p = 0.45$) nor mean specific production rate ($\mu$) ($n = 8$, $r = 0.247$, $p = 0.56$) was significantly related to mean fish abundance ($\theta$).

Throughout the study period, the mass ratio of zooplankton to phytoplankton varied temporally (Supplementary Fig. 4). Among treatments, the temporal mean of this ratio ($H^*/P^*$) was highest in the mid-shade treatment and lowest in the low-shade treatment in both ponds (Fig. 2(b)). However, $H^*/P^*$ was higher in pond 218 than in pond 217. A significant relationship was not detected between the $H^*/P^*$ and mean PAR in the water column (Fig. 2b; $n = 8$, $r = 0.155$, $p = 0.714$), mean fraction of edible phytoplankton ($\alpha_{edi}$) (Fig. 3a; $n = 8$, $r = 0.241$, $p = 0.565$), mean

seston carbon to phosphorus ratio ($\alpha_{nut}$) (Fig. 3b; $n = 8$, $r = -0.265$, $p = 0.523$), and mean specific production rate ($\mu$) (Fig. 3c; $n = 8$, $r = 0.081$, $p = 0.849$), whilst a significantly negative relationship was detected between the $H^*/P^*$ mass ratio and mean of fish abundance (CPUE) (Fig. 3d; $n = 8$, $r = -0.818$, $p = 0.013$).

We fitted $H^*/P^*$ by $\alpha_{edi}$, $\alpha_{nut}$, $\mu$, and $\theta$ among treatments in the two ponds using a multiple regression linear model. As fish were often not collected, we used $\theta = \text{CPUE} + 1$ as a relative measure of fish abundance. The variance inflation factors (VIFs) for these explanatory variables ranged from 1.05 to 2.38, indicating a low probability of multicollinearity among explanatory variables. An analysis with the generalized linear model showed that the model including all of these parameters had the lowest Akaike's Information criterion (Supplementary Table 3), indicating that it was the best model. The multiple regression analysis revealed that all four variables were significant: 95% confidence intervals (CI) were smaller or larger than zero, and explained 95% of variance in $H^*/P^*$ (Table 1). The regression coefficient was significantly less than zero for seston carbon to phosphorus ratio ($\alpha_{nut}$) while it did not significantly differ from one for edible phytoplankton frequency ($\alpha_{edi}$) and specific production rate ($\mu$), and was smaller than one but larger than zero for fish abundance ($\theta$). Because sample size (two ponds × four treatments) was limited relative to the number of parameters in the multiple regression, the results may not be reliable due to low statistical power. Therefore, we examined the effects of these parameters separately using the partial regression analysis, and found that all the partial correlation coefficients of these factors were statistically significant (Fig. 4), indicating that these explanatory variables affected the $H^*/P^*$ independently. Finally, to examine sensitivities of $H^*/P^*$ to changes in $\alpha_{edi}$, $\alpha_{nut}$, $\mu$, and $\theta$, we estimated standardized regression coefficients. The absolute value of the coefficients for $H^*/P^*$ was highest for $\theta$, followed by $\alpha_{nut}$ (Table 1).

## Discussion

Starting with the green world hypothesis proposed by Hairston Sr. et al.[12] and the counterpoint by Ehrlich and Birch[32], a number of studies have examined the effects of primary production and

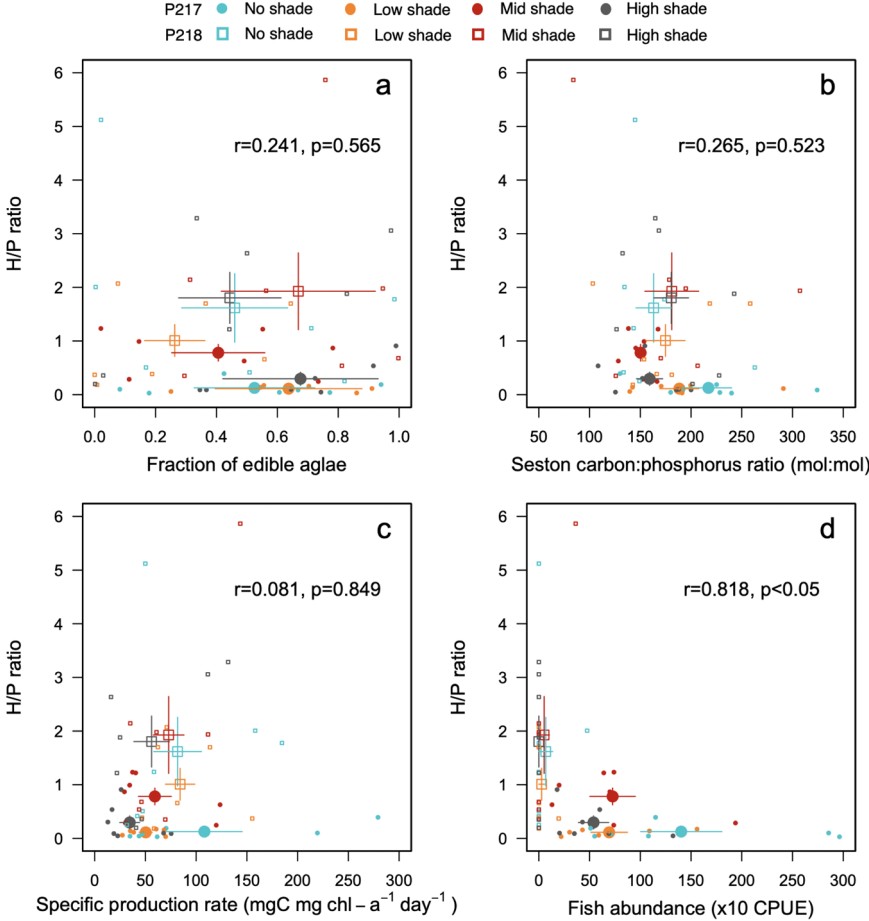

**Fig. 3 Relationships between H/P mass ratio and environmental variables.** H/P mass ratio plotted against edible phytoplankton fraction (**a**), seston carbon to phosphorus (C:P) ratio (**b**), specific production rate (**c**), and fish abundance (**d**) during the experiment in no-shade (blue), low-shade (orange), mid-shade (red), and high-shade (gray) treatments in pond 217 (circles) and 218 (squares). In each panel, small symbols denote values at each sampling date, and large symbols denote the mean values among the sampling dates. Bars denote standard errors on the means ($n = 7$ sampling date in each section). Correlation coefficients ($r$) with $p$ values between the mean values are inserted in each panel.

**Table 1 Results of multiple regression analysis.**

| Variables | Parameters in Eq. (9) | | Regression coefficient | 95% CI | Standardized regression coefficient |
|---|---|---|---|---|---|
| | Intercept | $\gamma$ | 29.31 | 19.30~38.09 | – |
| Seston C:P ratio | $\log(\alpha_{nut})$ | $\varepsilon_1$ | −7.07 | −9.21~−4.93 | −0.73 |
| Fraction of edible phytoplankton | $\log(\alpha_{edi})$ | $\varepsilon_2$ | 1.19 | 0.42~1.97 | 0.34 |
| Specific daily production rate | $\log(\mu)$ | $\varepsilon_3$ | 1.80 | 1.05~2.54 | 0.58 |
| Fish abundance | $\log(\theta)$ | $\varepsilon_4$ | 0.50 | 0.39~0.62 | 0.81 |

The regression coefficient for each independent variable in the multiple regression analysis ($n = 8$ treatments in two ponds, $R^2 = 0.95$, $p < 0.05$). 95% confidence intervals estimated by bootstrapping procedure ($n = 1999$), and standardized regression coefficients for each variable are shown.

predation[6,7,13,14,16,33], and those of anti-herbivory defense or edibility of producers[8–11,20] on herbivore abundance relative to producer abundance (the H/P ratio). Although the elemental stoichiometry of primary producers has often been considered as a factor that determines herbivore biomass[2,4,26], only a few studies have examined whether nutrient content can regulate herbivores relative to producer abundance using planktonic communities[34,35]. Moreover, to the best of our knowledge, no study has simultaneously examined the effects of these putative factors on the H/P ratio in nature, presumably because no theoretical framework has been developed for examining their effects in a comparable way. This is the first study to examine simultaneously the effects of those four factors on $H*/P*$ in a single

natural community. By fitting our observed data to a modified Lotka–Volterra-based model, we have shown that in addition to primary production and predation, which were repeatedly pointed out as fundamental factors affecting the community structure, edibility, and stoichiometry of primary producers also play pivotal roles in regulating $H*/P*$. In other words, this study showed that the importance of a producer's stoichiometry and edibility on herbivore abundance becomes apparent only if the effects of primary production and predation are simultaneously examined.

Our model, based on Lotka–Volterra equations, is derived from the equilibrium state of a system. In this study, both phytoplankton and zooplankton biomass changed markedly in all the treatments

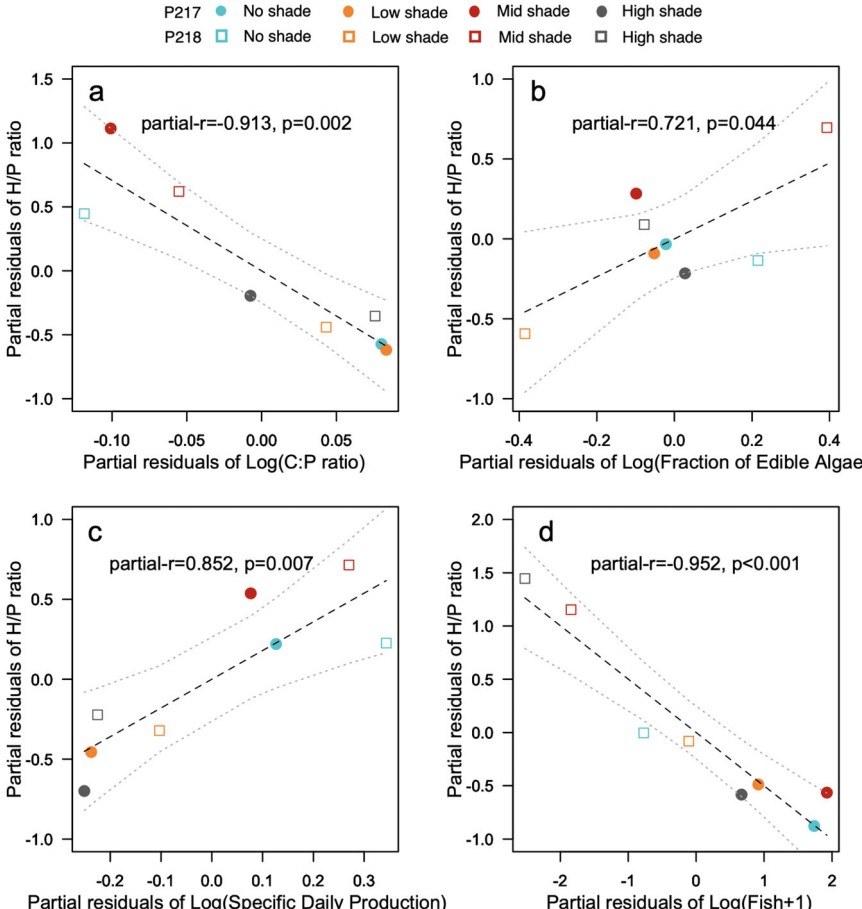

**Fig. 4 Partial effects of edible phytoplankton fraction, seston C:P ratio, specific production rate, and fish abundance on H/P mass ratio.** Partial regression leverage plots showing relationships between *H/P* mass ratio (the response variables), and log-transformed C:P ratio of seston (**a**), fraction of edible algae (**b**), specific daily production (**c**), and relative fish abundance (**d**) (the explanatory variables) without interfering effects from other explanatory variables. The vertical axis represents the partial residuals of *H/P* mass ratio, and the horizontal axis represents the partial residual of the specific explanatory variable. Dashed and dotted lines in each panel represent the partial regression line and its 95% confidence curves. Partial correlation coefficients with *p* values are also inserted in each panel. Data from four different treatments of pond 217 (circles) and 218 (squares) are denoted by different colors.

throughout the experiment, indicating that no community in this study reached equilibrium. However, theoretically, temporal means of herbivore and producer abundances for at least one oscillation cycle should coincide with the equilibrium abundances in the Lotka–Volterra model[25]. Theoretical and experimental studies have also shown that a single cycle of abundance occurred within <50 days in zooplankton–phytoplankton dynamics[36]. Thus, the present experimental run (85 days) was longer than at least one oscillation. In addition, we sampled at regular intervals. Therefore, the temporal mean values among samples would be close to equilibrium values in zooplankton–phytoplankton dynamics.

In our experiment, $H^*/P^*$ was most sensitive to changes in fish abundance among factors compared with the production, edibility, and stoichiometry of producers, suggesting that top–down control by zooplanktivorous fish is the greatest factor affecting $H^*/P^*$ in our planktonic community. In all sections (treatments) of pond 217, where fish were abundant, the density of large cladocerans in the zooplankton community was low, a result in accordance with the well-known obsrvation that planktivorous fish prey selectively on larger zooplankton species[13,30,33]. However, the high sensitivity of $H^*/P^*$ to changes in the fish abundance may reflect the possibility that the variation of fish abundance was more substantial than variation in other factors in our experimental setting. Thus, the present result does

not imply that compared with predation, other factors were less critical in regulating $H^*/P^*$ in nature.

Other than the direct effect of predation, several studies suggest that fish can indirectly affect zooplankton biomass by stimulating primary productivity through nutrient recycling[37]. However, in this study, the specific primary production rate ($\mu$) was not related to fish abundance, suggesting that the net impact of fish abundance on $H^*/P$ was largely attributable to a direct top–down force on zooplankton biomass rather than indirect bottom–up forcing through nutrient cycling.

A number of studies have argued that $H^*/P^*$ is regulated by the efficacy of the producer's anti-predator defense[8,10,11]. In this study, occurrence of phytoplankton, such as cyanobacteria, that might have contained compounds toxic to zooplankton[18,30] was limited. Therefore, we have focused on the physical defense traits of phytoplankton. As most herbivorous plankton cannot effi-ciently graze phytoplankton species with a cellular or colony size larger than 30 μm[30], enlargement of cellular or colony size can be viewed as a defense trait against herbivory[21]. Therefore, we examined effects of the edible phytoplankton fraction ($\alpha_{edi}$) on $H^*/P^*$. No significant relationship was detected between these. However, if we considered only the treatments in pond 218 where fish abundance was limited, $H^*/P^*$ tended to increase with this fraction. Indeed, $\alpha_{edi}$ was significantly related with the mass ratio

when other factors, such as fish abundance, were simultaneously considered in the multiple regression analysis. These results indicate that edibility, or a defense trait such as enlargement of cellular or colony size, indeed has a role in regulating $H^*/P^*$ in nature.

Other than the defense traits, nutritional value or nutrient content of producers have often been proposed as crucial factors determining the abundance of herbivores relative to that of producers[2,4,22,23]. In this study, cyanobacteria biomass was <20%, suggesting that a deficiency of polyunsaturated fatty acids and sterol was not a prime factor affecting the quality of phytoplankton food for zooplankton[27]. Cebrian[22] argued that a lower $H^*/P^*$ in terrestrial communities compared with aquatic communities is attributable to lower nitrogen and phosphorus contents relative to carbon in terrestrial producers. In this study, we focused on phosphorus as the main nutrient since phosphorus limitation of herbivore growth at an individual level has been repeatedly pointed out as an outcome of ecological stoichiometry[4,38]. Indeed, this study has shown that the seston carbon to phosphorus ratio was a significant factor affecting $H^*/P^*$ across communities with different taxonomic compositions of phytoplankton and zooplankton. This result is consistent with theoretical predictions of ecological stoichiometry, which states that the relative abundance of herbivores to producers changes depending on the stoichiometric mismatch between them[4].

In conclusion, the theoretical framework presented in this study showed that herbivore biomass relative to producer biomass is related significantly with the abundance of carnivores, producer stoichiometry, producer defense traits, and primary production rate in plankton communities. The experimental results support our hypothesis that these factors can simultaneously regulate community structure in nature. In this study, we considered the size and phosphorus stoichiometry of phytoplankton as the defense trait and nutritional quality of primary producers, respectively. However, it is also possible to consider the effects of other chemical and physical defenses such as secondary metabolites, toxins and thorns, and nutritional substances such as protein and essential fatty acid contents in Eq. (4). In our theoretical framework, we did not consider organisms' size or temperature, which would affect the biomass ratio of herbivores to producers through differences in their size- and temperature-specific metabolic rates[39]. Thus, caution is needed to apply the present model to communities for which size structures differ over several orders of magnitude. Under such conditions, our theoretical framework (Eqs. (4) and (9)) can incorporate multiple factors, so that application of the model to various terrestrial and aquatic communities is possible. Thus, the present theoretical framework may serve to generalize the relative importance among production rate, defense traits and stoichiometric nutrient content of producers, and predation in various ecosystems.

## Methods

### Experimental design
The experiment was carried out at two ponds (pond ID 217 and 218) located at the Cornell University Experimental Ponds Facility (CUEPF) in Ithaca, NY, USA (42°30'N, 76°26'W) during 4 June to 28 August 2016 (Fig. 1). At the Neimi Road Site (Unit 2) of CUEPF, 50 ponds were built in 1964 and have since been used for various field experiments in aquatic ecology[31,33,40]. Each pond has a 0.09 ha surface area (30 × 30 m) and is 1.5 m deep. Before the experiment, we pumped out the water as much as possible, removed as many macrophytes as possible by hand to equalize the environmental conditions of the ponds, installed vinyl-coated canvas curtains to divide each of the two ponds into four equal sections that were squares 15 m on a side, and then refilled the ponds with filtered (through a 1 mm mesh) using water from a reservoir source. The curtains were suspended from floats at the surface and held against the pond bottom with heavy metal chains and concrete weights. During this procedure, we could not remove planktivorous fish from the ponds so fish abundance was not manipulated. In each pond, we randomly assigned the four sections to one of four treatments: high-shade (64% shading), mid-shade (47% shading), low-shade (33% shading), or no-shade treatments (no shading). Shading in each treatment was made using different

number of opaque floating mats (6 m diameter; Solar-cell SunBlanket, Century Products, Inc., Georgia, USA)[31] (Fig. 1).

### Samplings
Phytoplankton and zooplankton were collected biweekly during the experiment. In each treatment (section) of the two ponds, 11-L water was collected in duplicate from the bottom to surface with the repeated deployment of a 2.2-L tube sampler (5 cm in diameter × 112 cm in length). These were used for measuring water chemistry and abundance of phytoplankton and small zooplankton (copepod nauplii and rotifers). In addition to these samples, crustacean plankton were collected by filtering 30-L of vertically integrated water from three different sites in each section with a 100 μm mesh net, and fixed with 99% ethanol for enumeration. During sampling, we measured vertical profiles of water temperature, DO concentration, conductivity and pH using a multiparameter probe (600XLM, YSI) at each section of both ponds. We also measured PAR using a spherical quantum sensor (LI-193; LiCor, Inc.) at 10-cm intervals from the surface to bottom and calculated extinction coefficients of PAR in the water.

### Chemical analyses and enumeration of plankton
In the laboratory, sestonic particles in the water, including phytoplankton, were concentrated onto precombusted GF/F (0.7 μm pore size) filters. Seston carbon was determined with a CN analyzer (model 2400; Perkin-Elmer, Inc.). Seston phosphorus concentrations were determined by the ascorbate-reduced molybdenum-blue method[41]. Chlorophyll a was extracted by 90% ethanol for 24 hours in the dark and quantified using a fluorometer (TD-700; Turner Designs, Inc.). For phytoplankton and small zooplankton samples, 250-mL and 500 mL, respectively, of the collected water were fixed with dilute Lugol's solution. These samples were settled by gravity for >24 hours and concentrated into 20 mL prior to analysis. For each phytoplankton taxon, the number of cells in 0.2–1.0 mL of the concentrated sample were counted and the sizes of 20–50 cells were measured for estimating the cell volume, which was made based on geometric shapes. For small zooplankton, we counted all individuals in 1 mL according to taxa with measurements of body length and width. For crustacean zooplankton, we concentrated the samples into 10 mL. For each of the crustacean taxa, we categorized size classes according to body length and counted individuals of each size category in 1 mL of the concentrated sample.

### Biomass estimation
We estimated carbon biomass of phytoplankton from the cell biovolume (μm³) using conversion factors presented by Menden-Deuer & Lessard[42]. Although large zooplankton individuals can graze on algae up to ~ 70 μm in size[33], most zooplankton including small cladocerans and copepods are known to graze efficiently on algae smaller than 30 μm in size[30]. Therefore, we estimated the fraction of phytoplankton smaller than 30 μm for the major axis of the cell or colony in the total phytoplankton biomass as edible fraction ($\alpha_{edi}$). Carbon biomass of cladocerans and copepods were determined using individual abundance and length-weight relationships[43] with a conversion factor of 0.48 g C per g DW. For rotifers, species-specific biovolume was estimated[44]. Then, carbon biomass of rotifers was estimated using individual abundance, specific biovolume and conversion factors of 0.024 g C per g WW except for Asplanchna and Keratella, which were determined by species-specific carbon weight for each of these taxa[45,46].

### Primary production rate
For estimating the daily production rate, we measured changes in $O_2$ concentration rate per unit of chlorophyll a (μg $O_2$ μg chl-$a^{-1}$ h$^{-1}$) using the light and dark bottle method[41] on 14 June, 11 July, and 22 August. Pond water from each pond section was poured into each of 16 100-ml DO bottles. An aliquot of the water was also collected for quantifying the chlorophyll a concentration as described above. In each treatment, two of these bottles were fixed immediately to quantify the initial DO concentration. Two of these bottles were wrapped in aluminum foil, incubated at 0.5 m depth and used for measuring community respiration rate. The remaining 12 bottles were incubated at various depths from the surface to bottom in the no-shade treatment section of pond 218. Simultaneously, we measured PAR at depths where the bottles were incubated. After six hours incubation from 10:00 to 16:00, we measured $O_2$ concentrations in all the DO bottles by the Winkler method. Then, according to Wetzel and Likens[41], chlorophyll a-specific photosynthetic rate (g C g chl-$a^{-1}$ min$^{-1}$) was calculated by assuming that C fixation and $O_2$ production occurred in a 1:1 stoichiometric ratio.

To estimate PAR during the bottle incubation, the light extinction coefficient (λ: Supplementary Fig. 1(d)) was calculated by fitting photon flux density (I) against depth (z) in the no-shade treatment section of pond 218 according to the following equation:

$$I(z) = I(0)\exp(-z). \qquad (10)$$

Mean PAR during the incubation was calculated using λ and temporal changes in I(0) that were estimated from ambient PAR in air ($PAR_{air}$). The $PAR_{air}$ was obtained from Guterman Research Center at Cornell University, near CUEPF, where $PAR_{air}$ was monitored every 2 minutes. Then, in each of the pond sections with different treatments in the two ponds, specific photosynthetic rates were plotted against the mean PAR and fitted to non-rectangular hyperbola models by the function nls() in R 3.2.1[47] to obtain the photosynthesis–PAR curve (Supplementary Fig. 8). Using the photosynthesis– PAR curve and daily PAR in the water column, we estimated chlorophyll a specific daily production rate. The daily

PAR in each pond section was derived from $PAR_{air}$, and $\lambda$ (Supplementary Fig. 2) at the time of sampling. To minimize specificity in light conditions at the measurement date, we estimated chlorophyll $a$ specific daily production rate using the temporal profile of PAR for 3 days before each sampling date. Then the average value for these 3 days was used in our analysis. On the date when photosynthetic rate was not measured, we used the photosynthesis–PAR curve obtained for the closest measured date. Because we used carbon biomass of phytoplankton for estimating the H/P mass ratio, we used chlorophyll $a$ specific daily production rate as a surrogate of the specific production rate ($\mu$) to avoid autocorrection between this and the H/P ratio.

**Fish abundance**. In each of the different treatment sections, fish were sampled using minnow traps with carp bait. The traps were placed 3–5 days before the regular plankton samplings in each section of the ponds. Then, we collected the traps at the plankton samplings and measured total wet weight of the collected fish (g) as CPUE. Note that, although shading may have affected the number of fish collected by the traps, CPUE can nevertheless be viewed as a measure of fish activity and, therefore, an indication of relative predation pressure[48]. Work with fish was permitted under Cornell University's Institutional Animal Care and Use Committee protocol 2016-0095.

**Statistics and reproducibility**. In each treatment ($n = 8$: four treatments × two ponds) of the field experiment, we used mean values from June 10 to August 27 ($n = 7$ sampling dates) for all variables in the following statistical analyses (Table S2). Relationships among phytoplankton and zooplankton biomasses, specific production rate and fish abundance were examined by correlation analysis. To test differences in phytoplankton and zooplankton community composition among the treatments and between the two ponds, PERMANOVA was performed by the adonis() function in R package "vegan"[49]. In this test, we used 999 permutations and the Euclidean distance both for phytoplankton and zooplankton communities as an index of dissimilarity in the community.

We used mean phytoplankton carbon biomass, zooplankton carbon biomass, fish abundance, specific production rate and fraction of edible phytoplankton ($n = 8$: four treatments × two ponds) for $P$, $H$, $\theta$, $\mu$, and $\alpha_{edi}$ in Eq. (9), respectively. As our traps often contained no fish, we used $\theta = CPUE + 1$. For $\alpha_{nut}$, we focused on phosphorus since freshwater limnetic ecosystems are primarily phosphorus limited[17,18] and as growth of zooplankton is affected by relative phosphorus in algae[26–29]. Specifically, we used the carbon to phosphorus ratio of seston as a surrogate for $\alpha_{nut}$ because this ratio has been generally used in consideration of ecological stoichiometry in freshwater[4]. Thus, we expected lower $H^*/P^*$ at larger values of seston carbon to phosphorus ratio. To examine effects of these explanatory variables on the $H/P$ ratio, a simple regression analysis was performed. Then, after checking multicollinearity among the explanatory variables by VIFs[50], we fitted these data to Eq. (9) using a lm function of R 3.2.1[47] followed by examination of Akaike's Information Criterion. In this analysis, 95% CIs of the regression coefficients were estimated using bootstrapping with a residual resampling procedure[51] and 1999 replicates. Because Eq. (9) indicates an a priori effect direction of a given variable, we estimated upper or lower one-tailed 95% CIs (100 and 5 percentiles) for the explanatory variables according to negative or positive effects predicted by Eq. (9). Effect sizes of these explanatory variables were assessed using standardized regression coefficients of the multiple regression. Finally, to examine whether effects of explanatory variables on the $H/P$ mass ratio were independent of each other and significant, we performed partial regression analysis with residual leverage plot according to Sall[52] using leveragePlots() in R package "car"[53].

**Reporting summary**. Further information on research design is available in the Nature Research Reporting Summary linked to this article.

## Data availability
Data used in this study are summarized in supplementary Table 2 and available at the Dryad repository https://doi.org/10.5061/dryad.p8cz8w9ms[54].

## Code availability
The codes used this study are available at the Dryad repository https://doi.org/10.5061/dryad.p8cz8w9ms[54].

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

## Acknowledgements

We thank N. Hamm and R. L. Johnson for managing the experimental ponds, L.R. Schaffner for helping with laboratory and fieldwork and M. Kyle for discussion. This project was supported by the Japan Society for the Promotion of Science (JSPS) Grant-in-Aid for Scientific Research (KAKENHI) 15H02642 to J.U., M.Y., I.K., H.D., and T.Y., 16H02522 and 20H03315 to J.U., and 16K18618, 16H04846, and 18H02509 to M.Y.

## Author contributions

J.U., T.K., and N.G.H. planned and designed the enclosure experiment. N.G.H. arranged use of CUEPF ponds for the experiment. T.K., J.U., K.T., X.Y., M.Y., I.K., H.D., T.Y., and N. G. H. contributed to the experimental setup and carried out the experiment. T.K., K.T., and X.Y. performed chemical analyses, enumerated plankton, and measured primary production rates. J.U. and M.Y performed theoretical modeling and J.U. and T.K. performed statistical analyses. J.U., T.K., M.Y., and N.G.H. wrote the draft and all authors contributed to the final manuscript.

## Competing interests

The authors declare no competing interests.
