## [Peer Review File · Communications Biology]

Reviewers' comments:

Reviewer #1 (Remarks to the Author):

GENERAL COMMENTS

This manuscript addresses a longstanding issue in community ecology - what controls the flow of energy and nutrients in foodwebs (producers or predators)? The issue had been polar for a while, and no study has rigorously tested both - this study does. As such, we are enthusiastic about the study. The authors lay out nicely in their introduction why they used the factors in their study (primary production, defense traits of producers, nutrient content of producers, and predation rate) in determining H/P ratio. Their framework was logically constructed and parameterized. Their methods were sufficiently explained to reproduce the work, and the analyses were appropriate to determine their objectives. In addition, their conclusions were not overreaching and match the analyses done/results presented.

While we did not have any major/fatal concerns, we did worry about replication, and the key differences in the two replicates. Nevertheless, we felt the authors were well aware of this, and their inferences are quite parsimonious. We would like some clarification on sampling effort for the various parameters. Were efforts to measure fish biomass, phytoplankton biomass, and zooplankton biomass comparable and synchronous? They are being used as estimates of parameters to get which has most effect on H/P—so sampling intensity/synchrony could bias results, particularly because the system was not at equilibrium (as the authors acknowledge; and reason the system was “near” it). Given the low replication, the authors could not analyze the ponds separately (which would’ve been ideal). Perhaps one way to get at this may be to qualitatively analyze the ponds (no need for P values; simply compare responses in the two ponds visually).

We have done our best here to provide editorial support, but we recommend that the manuscript go through another round of grammatical edits (preferably by an anglophone) before resubmission.

LINE-SPECIFIC COMMENTS

Introduction

L68 – somewhere in this paragraph, it would be great to give a flavor for the magnitude of variation in H/P among communities and/or seasons within a community. Some way to emphasize such work has substantial practical implications, and prime the reader to such heterogeneity (evident in the results).

L84 – primary (I found myself thinking about prime factorization!)

L92 – In this study, we used classical Lotka-Volterra equations to develop a framework that simultaneously assesses the effects of...on the H/P ratio.

L95 – We tested the model using data from communities inhabiting experimental ponds. A bottom up factor (light) was manipulated by shading, while the top-down factor (fish predation) was quantified regularly.

Line 136-change ‘few amounts of carnivores’ to ‘low abundance of carnivores’

Line 139- ‘An increase’ rather than ‘Increase’ and similarly ‘a decrease’ in the same line

Line 143-‘decreases’ rather than ‘decrease’

Line 159-‘due to gape limitation’ rather than ‘due to a gape limitation’

Theoretical framework

We have checked the math and could follow the derivations based on the information given.

Results

Line 191- add ‘is’ for ‘and is 1.5m deep.’

Line 192- ‘divided’ rather than ‘divide’ to keep verb tense consistent

Line 229—fix citation

L268 - ...all four variables (delete "of these")

Lines 278-279- this is an awkward phrase, consider 'indicating that these explanatory variables affected the H*/P* mass ratio independently'. Significance already indicates the factors are potentially important.

Discussion

Line 284-fix citation.

L290 – herbivores

L290 – something is off. Is the point that it hasn't been tested in mesocosms with natural communities (as Urabe et al. 2002 did), but has been in the lab? Please consider rephrasing. If the emphasis is on natural communities, then a few words about key lab artifacts would enable readers to better appreciate this and other mesocosm studies.

L291 – this sentence is confusing. ...knowledge, no study has simultaneously examined the effects of bottom-up and top-down factors on the H/P ratio.

Line 292- remove 'on' from 'simultaneously affect on the herbivore...'

L294 - ...to simultaneously examine...

L297 – this is a key take home that isn't developed in the Intro sufficiently. The importance of such other factors (e.g., stoichiometry, edibility) become apparent only if top-down and bottom-up factors are simultaneously examined!

L341 – don't get the emphasis on aquatic communities – is it needed? Else, you will have to discuss more explicitly the role of secondary metabolites in terra.

L351 – cite the original GRH paper, and a recent review?

L353 – This result is consistent with theoretical predictions of ecological stoichiometry, which states...

L364 – keep it general in the last paragraph. ...chemical and physical defenses in eq. (4).

L368 – True, but do we expect major size effects in aquatic foodwebs, where big things eat small things? So a size correction may not be of major use here compared to terrestrial? One option is to correct for T and size like the MTE folks do?

Methods

2.2 L tube sampler does not have specific info, neither did the mesh net

H/P leverage is not explained (axes in figure 4)

Declarations

Line 538. Typos: peer and download

Figures and tables.

Typo in table 1

Figure 4. H/P Ratio leverage...

Fig S9 – killifish (not killyfish)

Reviewer #2 (Remarks to the Author):

In this manuscript, the authors propose a simple framework to examine the structure of ecological communities, which is based on Lotka-Volterra equations describing the population dynamics of primary producers and herbivores. Specifically, this framework is used to analyze the biomass ratio of herbivores to primary producers, which is assumed to be potentially shaped by four ecological factors: primary production, nutritional quality and edibility of primary producers, and predation by carnivores. The proposed framework is tested on natural plankton communities, and indicates that the biomass ratio of herbivores to primary producers in such communities is significantly affected by all those four ecological factors.

Overall, I find that this manuscript is well presented and clearly written, and that the proposed

framework is suitable to investigate not only the structure of plankton communities, but also that of other aquatic and terrestrial communities. This manuscript will potentially draw the attention of a broad readership interested in community ecology, aquatic ecology and theoretical ecology, and therefore matches the scope of the journal to which it was submitted.

However, I think that the Introduction and Results sections still need some improvement, in order to enhance the quality and significance of the paper. In particular, I missed an explanation why the biomass ratio of herbivores to primary producers is of 'fundamental importance,' as stated by the authors in the Introduction. In my view, such an explanation will be necessary to understand the important role of the biomass ratio of herbivores to primary producers in ecological communities. Furthermore, I found a few inconsistencies in some figures that I would like to see addressed in the Results section.

Major comments

1. In the Introduction, you state that the biomass ratio of herbivores to primary producers is of 'fundamental importance' (line 69). Please elaborate briefly on this statement, so that readers may better understand the important role of the biomass ratio of herbivores to primary producers in ecological communities.
2. Although the Abstract is well structured and to the point, I find its last sentence a bit confusing and underwhelming (lines 55-57). In my view, this sentence should be revised to clearly indicate that, given the wide scope of your framework, it will prove useful to examine the factors shaping several types of terrestrial and aquatic communities.
3. In figures S1, S3, S4 and S7, data points obtained from pond 218 should be plotted according to the legends as empty squares, and not as empty circles.
4. Table S2 showing temporal means and standard errors of plankton biomass, primary production rate, fish abundance, and sestonic elemental ratios is not referred to in the main text. Please either explain these results in the main text, or remove Table S2 from the manuscript.

Minor comments

1. Throughout the Introduction, you refer to 'bottom-up forces' and 'top-down forces' that shape the biomass ratio of herbivores to primary producers. Although I understand your terminology, it will perhaps be clearer and more exact to use the terms 'bottom-up control' and 'top-down control.'
2. Lines 90-91 can be removed, because this text essentially repeats what is previously written in lines 74-76.
3. In equation (2), make sure that all symbols are italicized, and that the functional response in herbivores always reads ' $f(P^*)$ '.
4. In line 176, ' H^*/P^*P ' should read ' H^*/P^* '.
5. In lines 229-230, '(24Lampert and Sommer 2007)' should read '(24)'.
6. In line 256, 'Fig. 2(a;)' should read 'Fig. 2(b)'.
7. In line 267, '(Table S2)' should read '(Table S1)'.

8. In line 284, '10Hairston et al. (10)' should read 'Hairston et al. (10)'.
9. In line 331, 'we focused physical defense' should read 'we focused on the physical defense'.
10. In line 357, 'With the theoretical framework, this study' should read 'The theoretical framework presented in this study'.
11. In lines 359-360, 'primary production rate in the plankton communities' should read 'primary production rate in plankton communities'.
12. In line 361, 'regulate the community structures' should read 'regulate community structure'.
13. In lines 365-366, 'In the theoretical framework' should read 'In our theoretical framework'.
14. In line 396, 'Samplings' should read 'Sampling'.
15. In line 501, 'analyses' should read 'analysis'.
16. In line 523, 'planed' should read 'planned'.
17. In line 538, 'pear review' should read 'peer review'.
18. In Table 1, the variable 'Ffraction of edible phytoplankton' should read 'Fraction of edible phytoplankton'.
19. In the caption of Figure S7 (line 110), 'N:P rations' should read 'N:P ratio'.
20. In the caption of Figure S8 (lines 118-119), please remove the legends (C), (L), (M) and (H) for no shade, low shade, mid shade and high shade sections, respectively, because the plots do not refer to such legends.
21. In the legend of Figure S9, 'Killyfish' should read 'Killifish'.

Reviewer #3 (Remarks to the Author):

Review: A unified framework for understanding biomass ratio of herbivores to producers with a field test of plankton

This manuscript provides a theoretical approach for understanding the relative effects of different factors on the biomass ratio between herbivores and producers (H:P). The analysis is interesting and provides what I think is a new way to align theoretical predictions with experimental observations. The experimental results are also suggestive (although not conclusive, given the small sample size) of the validity of the derivation. I have several suggestions below for clarifying the presentation.

Line 74: Better to say something like "...quantifying how these factors act together to affect H:P has been difficult..."

Line 97: Would be good here to link back to the factors described above, by stating that primary production rate was manipulated by altering light availability.

Line 106: The derivation would be simpler if the authors just start with the equilibrium solution to Lotka-Volterra, rather than going through the temporal form of the equation. Most readers should know this equilibrium solution, and then it's just a matter of describing how the parameters translate to the production, mortality rate, conversion efficiency, and grazable fraction.

Line 120: I'm not sure why $f(P)$ and $f(P^*)$ are retained in this equation. It would seem that they cancel each other out.

Line 128: There is somewhat of a leap in logic here to get to beta as interpreted by the authors because g is specific primary productivity, and beta is interpreted in terms of standing stock of inedible plankton. Some more explanation here would be good.

Line 136: "few amounts of carnivores": might be clearer to say "communities with relatively low carnivore abundance would have a correspondingly low value of m..."

Line 144: I think the assumptions used here to link the factors in the Lotka-Volterra to parameters that were measured in the experiment need to be described in much greater detail. For example, on Line 147, the assumption is that most of the variations in conversion efficiency can be attributed to differences in C:P. This assumption may be true, but more references are needed to justify the assumption, and other possible factors should be described (perhaps partly in the discussion). A similar expansion the description should accompany the other parameters. Also, in this section, it would be helpful to provide a table to crosswalk the final regression parameters (e.g. epsilon1) with the generic factors (e.g., conversion efficiency) with the measurement (e.g., C:P). Without such a table, the meaning of the different regression parameters is difficult to follow in the rest of the manuscript.

Line 214: It's not clear to me how the effects of initial conditions are removed by considering site means. Instead, it seems that analysis of site means just controls for the effects of temporal variability on the analysis.

Line 230: Here, different statistics are reported regarding the proportion of edible phytoplankton and proportion cyanobacteria. Many researchers have assumed that these two measurements are similar (i.e., cyanobacteria are generally inedible), but it seems here that they are different. Were the two measurements correlated?

Line 260: Is mean fish abundance $\log(x+1)$ transformed? If so, that should be stated here.

Line 276: Partial regression analysis doesn't really address the issue of low sample power because the effects of all the other predictors are still included in the adjusted value of the response. The small sample size is an issue for interpreting the results, but from Fig 3, it seems that the main variable that drives differences in H/P is fish abundance. Perhaps an effective way to display the data is to correct H/P for the effects of fish abundance using a simple linear regression, and plot this corrected H/P (i.e., the residuals of the regression against fish abundance) versus the other variables? This is a partial regression approach, but only using the one dominant variable to correct, rather than all other covariates. I suspect that the relationships with the other covariates will become more visible after correcting for just fish abundance and lend support to the MLR results. (Incidentally, I thought I might try this out, but the data were not available to reviewers in the URL provided by the authors).

Line 305: Based on this estimate of cycle length, the duration of the experiment may have captured slightly more than 1 cycle. Is this correct? It might be good to state this explicitly and argue that by sampling the cycle on a regular basis, the measurements approximate the mean conditions.

Response to the reviewers' comments and suggestions:

Note: our reply to the comments are shown in blue colors with the line number in the revised manuscript.

Reviewer #1 (Remarks to the Author):

GENERAL COMMENTS

This manuscript addresses a longstanding issue in community ecology - what controls the flow of energy and nutrients in foodwebs (producers or predators)? The issue had been polar for a while, and no study has rigorously tested both - this study does. As such, we are enthusiastic about the study. The authors lay out nicely in their introduction why they used the factors in their study (primary production, defense traits of producers, nutrient content of producers, and predation rate) in determining H/P ratio. Their framework was logically constructed and parameterized. Their methods were sufficiently explained to reproduce the work, and the analyses were appropriate to determine their objectives. In addition, their conclusions were not overreaching and match the analyses done/results presented.

[Reply] We deeply appreciate invaluable comments and suggestions with editorial advice for improving our English presentation. We have revised our manuscript according to these comments, as shown below. We believe that we have appropriately and satisfactorily incorporated these comments into the revision.

While we did not have any major/fatal concerns, we did worry about replication, and the key differences in the two replicates. Nevertheless, we felt the authors were well aware of this, and their inferences are quite parsimonious. We would like some clarification on sampling effort for the various parameters. Were efforts to measure fish biomass, phytoplankton biomass, and zooplankton biomass comparable and synchronous? They are being used as estimates of parameters to get which has most effect on H/P—so sampling intensity/synchrony could bias results, particularly because the system was not at equilibrium (as the authors acknowledge; and reason the system was “near” it).

[Reply] In the experiment, we collected phytoplankton and zooplankton on the same day. We also collected fish when plankters were sampled: we set fish traps 2-5 days before the plankton sampling, and fish were collected at the time of plankton sampling. We have stated these points in the method section (L404-407 & L486-488). Thus, sampling intensity and intervals were the same for plankton and fish.

Given the low replication, the authors could not analyze the ponds separately (which would've been ideal). Perhaps one way to get at this may be to qualitatively analyze the ponds (no need for P values; simply compare responses in the two ponds visually).

[Reply] We appreciate this advice. We used different symbols for P217 (closed symbols) and P218 (open symbols) in all the figures to compare the results between the two ponds. As suggested, we did not compare statistically between the two ponds. However, readers will easily compare differences or similarities in the data ranges between these ponds from the different symbols on the figures.

We have done our best here to provide editorial support, but we recommend that the manuscript go through another round of grammatical edits (preferably by an anglophone) before resubmission.

[Reply] We appreciate this support and are certainly happy to have a check of English usage. In the revision, we have done our best to revise English, and one of the authors (NH), who is a native speaker, has carefully checked over the manuscript.

LINE-SPECIFIC COMMENTS

Introduction

L68 – somewhere in this paragraph, it would be great to give a flavor for the magnitude of variation in H/P among communities and/or seasons within a community. Some way to emphasize such work has substantial practical implications, and prime the reader to such heterogeneity (evident in the results).

[Reply] L68-69: We have added a known range of H/P biomass ratios with some citations.

L84 – primary (I found myself thinking about prime factorization!)

[Reply] L86: We have changed “prime” to primary.

L92 – In this study, we used classical Lotka-Volterra equations to develop a framework that simultaneously assesses the effects of...on the H/P ratio.

[Reply] L94: We have revised this sentence according to the comment.

L95 – We tested the model using data from communities inhabiting experimental ponds. A bottom up factor (light) was manipulated by shading, while the top-down factor (fish predation) was quantified regularly.

[Reply] L97-99: we have revised this sentence according to the comment.

Line 136-change 'few amounts of carnivores' to 'low abundance of carnivores'

[Reply] L137: We have changed to 'low carnivore abundance'.

Line 139- 'An increase' rather than 'Increase' and similarly 'a decrease' in the same line

[Reply] L140: We have revised these accordingly.

Line 143-'decreases' rather than 'decrease'

[Reply] L144: We have changed to 'decreases'.

Line 159-'due to gape limitation' rather than 'due to a gape limitation'

[Reply] L165-166: We have revised these accordingly.

Theoretical framework

We have checked the math and could follow the derivations based on the information given.

[Reply] Thank you for checking these.

Results

Line 191- add 'is' for 'and is 1.5m deep.'

[Reply] L195: We have inserted "is".

Line 192- 'divided' rather than 'divide' to keep verb tense consistent

[Reply] L196: We have unified the tense.

Line 229—fix citation

[Reply] L233: We have revised it.

L268 - ...all four variables (delete "of these")

[Reply] L272: We have deleted "of these" from this sentence.

Lines 278-279- this is an awkward phrase, consider 'indicating that these explanatory variables affected the H*/P* mass ratio independently'. Significance already indicates the factors are potentially important.

[Reply] L282: We have revised this sentence accordingly.

Discussion

Line 284-fix citation.

[Reply] L287: We have fixed citation numbers.

L290 – herbivores

[Reply] L293: we have corrected it.

L290 – something is off. Is the point that it hasn't been tested in mesocosms with natural communities (as Urabe et al. 2002 did), but has been in the lab? Please consider rephrasing. If the emphasis is on natural communities, then a few words about key lab artifacts would enable readers to better appreciate this and other mesocosm studies.

[Reply] L292-294: We have rephrased this sentence with the addition of a more recent study (citation #35) to show that the number of studies examining these effects using natural communities is limited.

L291 – this sentence is confusing. ...knowledge, no study has simultaneously examined the effects of bottom-up and top-down factors on the H/P ratio.

[Reply] L294-296: We have revised this sentence.

Line 292- remove 'on' from 'simultaneously affect on the herbivore...'

[Reply] L294-296 We have revised this sentence as above.

L294 - ...to simultaneously examine...

[Reply] L298: We have revised it.

L297 – this is a key take home that isn't developed in the Intro sufficiently. The importance of such other factors (e.g., stoichiometry, edibility) become apparent only if top-down and bottom-up factors are simultaneously examined!

[Reply] L302-305: We appreciate this comment. To make this point clear, we have added the following sentence:

“In other words, this study showed that the importance of a producer's stoichiometry and edibility on herbivore abundance becomes apparent only if the effects of primary production and predation are simultaneously examined..”

L341 – don't get the emphasis on aquatic communities – is it needed? Else, you will have to discuss more explicitly the role of secondary metabolites in terra.

[Reply] L345-347: We have removed “even in aquatic community” from this sentence and made it a more general statement. We have added “secondary metabolite” to L371 to argue the generality of the present approach.

L351 – cite the original GRH paper, and a recent review?

[Reply] L357: We have cited that paper (citation #38).

L353 – This result is consistent with theoretical predictions of ecological stoichiometry, which states...

[Reply] L359-360: We have rephrased this sentence as suggested

L364 – keep it general in the last paragraph. ...chemical and physical defenses in eq. (4).

[Reply] L370-372: To make this general argument clear, we have stated some examples in this sentence.

L368 – True, but do we expect major size effects in aquatic foodwebs, where big things eat small things? So a size correction may not be of major use here compared to terrestrial? One option is to correct for T and size like the MTE folks do?

[Reply] Thank you for this comment. Since metabolic rate is highly dependent on temperature and body size, we suggested these factors as possible subjects that should be examined in the future (L372-375). Actually, body size may not be a matter in the aquatic food web in lakes and ponds. But in terrestrial habitats, herbivore body size ranges by four orders of magnitude (mm size to several meter size). Thus, when we pooled these herbivores as a single component, we may need to make a body size correction, as pointed by Enquist et al. (2015) (citation #39). However, since this argument is not fundamental, we are happy to delete this sentence if the editor recommends doing so.

Methods

2.2 L tube sampler does not have specific info, neither did the mesh net
H/P leverage is not explained (axes in figure 4)

[Reply] We have inserted the size of the sampler on L 406. This sampler has no mesh since it collects whole water. We have revised explanations of x- and y-axis in Figure 4

with some modification of this figure's legend (L707-710).

Declarations

Line 538. Typos: peer and download

[Reply] L548: we have corrected these.

Figures and tables.

Typo in table 1

[Reply] We have corrected "Ffraction" to "Fraction" in Table 1.

Figure 4. H/P Ratio leverage...

[Reply] We have revised explanations of X- and Y- axes in Fig 4 to make these more understandable.

Fig S9 – killifish (not killyfish)

[Reply] Fig S9 We have corrected it.

Reviewer #2 (Remarks to the Author):

In this manuscript, the authors propose a simple framework to examine the structure of ecological communities, which is based on Lotka-Volterra equations describing the population dynamics of primary producers and herbivores. Specifically, this framework is used to analyze the biomass ratio of herbivores to primary producers, which is assumed to be potentially shaped by four ecological factors: primary production, nutritional quality and edibility of primary producers, and predation by carnivores. The proposed framework is tested on natural plankton communities, and indicates that the biomass ratio of herbivores to primary producers in such communities is significantly affected by all those four ecological factors.

Overall, I find that this manuscript is well presented and clearly written, and that the proposed framework is suitable to investigate not only the structure of plankton communities, but also that of other aquatic and terrestrial communities. This manuscript will potentially draw the attention of a broad readership interested in community ecology, aquatic ecology and theoretical ecology, and therefore matches the scope of the journal to which it was submitted.

[Reply] We are delighted to have these invaluable comments and suggestions to make our paper more robust. We believe that we have now appropriately revised the manuscript according to the comments and suggestions. Below we state how we have incorporated these comments.

However, I think that the Introduction and Results sections still need some improvement, in order to enhance the quality and significance of the paper. In particular, I missed an explanation why the biomass ratio of herbivores to primary producers is of ‘fundamental importance,’ as stated by the authors in the Introduction. In my view, such an explanation will be necessary to understand the important role of the biomass ratio of herbivores to primary producers in ecological communities. Furthermore, I found a few inconsistencies in some figures that I would like to see addressed in the Results section.

[Reply] L68-73: According to this comment, we have revised the first half of the 1st paragraph in the Introduction and stated the range of the H/P mass ratio and why the H/P mass ratio is ecologically important.

Major comments

1. In the Introduction, you state that the biomass ratio of herbivores to primary producers is of ‘fundamental importance’ (line 69). Please elaborate briefly on this statement, so that readers may better understand the important role of the biomass ratio of herbivores to primary producers in ecological communities.

[Reply] L69-71: we have briefly stated why the H/P ratio is ecological important, instead of simply asserting its “fundamental importance”.

2. Although the Abstract is well structured and to the point, I find its last sentence a bit confusing and underwhelming (lines 55-57). In my view, this sentence should be revised to clearly indicate that, given the wide scope of your framework, it will prove useful to examine the factors shaping several types of terrestrial and aquatic communities.

[Reply] L57: Due to the word number limitation, we could not change the abstract by adding to it. However, according to this suggestion, we have slightly revised it to make the implications of this study clearer.

3. In figures S1, S3, S4 and S7, data points obtained from pond 218 should be plotted according to the legends as empty squares, and not as empty circles.

[Reply] Fig. S1, S3, S4 and S7. We have corrected the plot symbols of Pond 218 to empty squares in these figures.

4. Table S2 showing temporal means and standard errors of plankton biomass, primary production rate, fish abundance, and sestonic elemental ratios is not referred to in the main text. Please either explain these results in the main text, or remove Table S2 from the manuscript.

[Reply] We have cited this table (Table S2 in the revision) on L221.

Minor comments

1. Throughout the Introduction, you refer to ‘bottom-up forces’ and ‘top-down forces’ that shape the biomass ratio of herbivores to primary producers. Although I understand your terminology, it will perhaps be clearer and more exact to use the terms ‘bottom-up control’ and ‘top-down control.’

[Reply] We have revised to “control” instead of “force” (L73, 79, 80 and 100).

2. Lines 90-91 can be removed, because this text essentially repeats what is previously written in lines 74-76.

[Reply] L92-93: We have rephrased this sentence since our intention (nuance) in this sentence is slightly different from those on L74-76.

3. In equation (2), make sure that all symbols are italicized, and that the functional response in herbivores always reads ‘ $f(P^*)$ ’.

[Reply] L121: We have corrected these.

4. In line 176, ‘ H^*/P^*P ’ should read ‘ H^*/P^* ’.

[Reply] L182: We have corrected it.

5. In lines 229-230, ‘(24Lampert and Sommer 2007)’ should read ‘(24)’.

[Reply] L233: we have corrected the reference and have removed “Lampert and Sommer 2007”.

6. In line 256, ‘Fig. 2(a;)’ should read ‘Fig. 2(b)’.

[Reply] L260: We have corrected it.

7. In line 267, '(Table S2)' should read '(Table S1)'.

[Reply] L271: We have revised it. Now it is Table S3 in this revision.

8. In line 284, '10Hairston et al. (10)' should read 'Hairston et al. (10)'.

[Reply] L287: we have corrected it.

9. In line 331, 'we focused physical defense' should read 'we focused on the physical defense'.

[Reply] L337: We have revised it.

10. In line 357, 'With the theoretical framework, this study' should read 'The theoretical framework presented in this study'.

[Reply] L364: We have revised this sentence according to the suggestion.

11. In lines 359-360, 'primary production rate in the plankton communities' should read 'primary production rate in plankton communities'.

[Reply] L364-367: We have revised it.

12. In line 361, 'regulate the community structures' should read 'regulate community structure'.

[Reply] L367-368: We have revised it.

13. In lines 365-366, 'In the theoretical framework' should read 'In our theoretical framework'.

[Reply] L372: We have revised it

14. In line 396, 'Samplings' should read 'Sampling'.

[Reply] L403: We have changed to "Sampling".

15. In line 501, 'analyses' should read 'analysis'.

[Reply] L510: We have corrected it.

16. In line 523, 'planed' should read 'planned'.

[Reply] L533: We have corrected it.

17. In line 538, 'pear review' should read 'peer review'.

[Reply] L548: We have corrected it.

18. In Table 1, the variable 'Ffraction of edible phytoplankton' should read 'Fraction of edible phytoplankton'.

[Reply] Table 1 We have corrected it.

19. In the caption of Figure S7 (line 110), 'N:P rations' should read 'N:P ratio'.

[Reply] Figure S7 We have corrected it (L767).

20. In the caption of Figure S8 (lines 118-119), please remove the legends (C), (L), (M) and (H) for no shade, low shade, mid shade and high shade sections, respectively, because the plots do not refer to such legends.

[Reply] Figure S8 we have made this change (L770-773).

21. In the legend of Figure S9, 'Killyfish' should read 'Killifish'.

[Reply] Figure S9 we have corrected it.

Reviewer #3 (Remarks to the Author):

Review: A unified framework for understanding biomass ratio of herbivores to producers with a field test of plankton

This manuscript provides a theoretical approach for understanding the relative effects of different factors on the biomass ratio between herbivores and producers (H:P). The analysis is interesting and provides what I think is a new way to align theoretical predictions with experimental observations. The experimental results are also suggestive (although not conclusive, given the small sample size) of the validity of the derivation. I have several suggestions below for clarifying the presentation.

[Reply] Thank you for your invaluable comments and suggestions. Below, we state how we have revised the manuscript according to these comments. We believe that the revision has appropriately incorporated the comments.

Line 74: Better to say something like "...quantifying how these factors act together to affect H:P has been difficult..."

[Reply] L75-76: We have revised this sentence according to the suggestion.

Line 97: Would be good here to link back to the factors described above, by stating that primary production rate was manipulated by altering light availability.

[Reply] L98-99: We have revised this sentence according to this suggestion.

Line 106: The derivation would be simpler if the authors just start with the equilibrium solution to Lotka-Volterra, rather than going through the temporal form of the equation. Most readers should know this equilibrium solution, and then it's just a matter of describing how the parameters translate to the production, mortality rate, conversion efficiency, and grazable fraction.

[Reply] L105: We appreciate this comment. Certainly, this framework can start from the equilibrium equations. However, a theoretical ecologist who critically read the early version of the manuscript suggested to us that it is a better to start from the original differential equations since these make it clear that this formation has been developed based on the Lotka-Volterra equations. Therefore, we have not changed this part. However, we will shorten this part according to this suggestion if the editor recommends doing so.

Line 120: I'm not sure why $f(P)$ and $f(P^*)$ are retained in this equation. It would seem that they cancel each other out.

[Reply] L120: We appreciate this comment. First of all, we have corrected the first $f(P)$ in this equation to $f(P^*)$. So, $f(p^*)$ appears two times in this equation and can be canceled as pointed out by the reviewer. However, this equation becomes more understandable if we denote implicitly that $H^* = [g(P^*) - x] / f(P^*)$ and $P^* = \{m / [k f(P^*)]\}$. Therefore, we have retained these and not canceled them in this equation.

Line 128: There is somewhat of a leap in logic here to get to beta as interpreted by the authors because g is specific primary productivity, and beta is interpreted in terms of standing stock of inedible plankton. Some more explanation here would be good.

[Reply] L128-132: To make our logic clearer, we have inserted sentences explaining the implications of beta and its rationale.

Line 136: "few amounts of carnivores": might be clearer to say "communities with relatively low carnivore abundance would have a correspondingly low value of m ..."

[Reply] L137: We have revised this sentence according to the suggestion.

Line 144: I think the assumptions used here to link the factors in the Lotka-Volterra to parameters that were measured in the experiment need to be described in much greater detail. For example, on Line 147, the assumption is that most of the variations in conversion efficiency can be attributed to differences in C:P. This assumption may be true, but more references are needed to justify the assumption, and other possible factors should be described (perhaps partly in the discussion). A similar expansion the description should accompany the other parameters.

[Reply] L148-153: In the experiment, we focused on how the relative phosphorus content in the primary producers is detrimental for the herbivores' growth efficiency. Therefore, as suggested, we have briefly argued the rationale for justifying our focus with some additional citations (#26-29). It is correct that some other nutrient substances also can affect the growth rate and efficiency, however, statements on these potential substances here may confuse readers by out of focus. Therefore, we have not mentioned them here. Instead, we have drawn attention to these other possible limiting factors in the conclusion of the manuscript (L368-372)

Also, in this section, it would be helpful to provide a table to crosswalk the final regression parameters (e.g. ϵ_1) with the generic factors (e.g., conversion efficiency) with the measurement (e.g., C:P). Without such a table, the meaning of the different regression parameters is difficult to follow in the rest of the manuscript.

[Reply] We have added a new table showing a list of variables (Table S1).

Line 214: It's not clear to me how the effects of initial conditions are removed by considering site means. Instead, it seems that analysis of site means just controls for the effects of temporal variability on the analysis.

[Reply] L218-221 We have removed data obtained at day 0 (at the start of experiment) to avoid possible artifacts that may have occurred due to activities while the experiment was being setting up (such as resuspension of sediments).

Line 230: Here, different statistics are reported regarding the proportion of edible phytoplankton and proportion cyanobacteria. Many researchers have assumed that these two measurements are similar (i.e., cyanobacteria are generally inedible), but it seems here that they are different. Were the two measurements correlated?

[Reply] No, they were not correlated. Other than large colonial cyanobacteria like *Microcystis*, only solitary (single cells) and small colonial cyanobacteria (<10um)

occurred. However, in total, these cyanobacteria occupied at most 20% (L233), generally < 20%, of the algal biomass, as shown in Fig S6.

Line 260: Is mean fish abundance $\log(x+1)$ transformed? If so, that should be stated here.

[Reply] L264: The correlation analysis was made without any transformation. To fit these data to eq. 9, we did a log-transformation. However, since 0 values were included in the fish abundance data, we applied a $\log(\text{CPUE}+1)$ transformation. We have explained this on L 266-267 & L503-504. We have explained CPUE where it first appears in the manuscript (L246).

Line 276: Partial regression analysis doesn't really address the issue of low sample power because the effects of all the other predictors are still included in the adjusted value of the response. The small sample size is an issue for interpreting the results, but from Fig 3, it seems that the main variable that drives differences in H/P is fish abundance. Perhaps an effective way to display the data is to correct H/P for the effects of fish abundance using a simple linear regression, and plot this corrected H/P (i.e., the residuals of the regression against fish abundance) versus the other variables? This is a partial regression approach, but only using the one dominant variable to correct, rather than all other covariates. I suspect that the relationships with the other covariates will become more visible after correcting for just fish abundance and lend support to the MLR results. (Incidentally, I thought I might try this out, but the data were not available to reviewers in the URL provided by the authors).

[Reply] We agree that the partial regression analysis does not solve the issue of low sample numbers. However, as suggested, we used it to show how our data are nicely scattered in the values range. We also agree that it is a good idea to plot the residuals of the regression of H/P ratio vs fish abundance (and thus regressed) against each of the other explanatory variables. In this case, however, we repeat the simple regression analysis three times using the shared data. Since these regression analyses are independent procedures, we would need carry out a Bonferroni correction to make statistical tests. Moreover, to repeat the same analysis with shared data is statistically redundant. Since we see no fundamental reason to repeat the simple regression analyses redundantly, we made this analysis in the frame of a single procedure of the partial regression analysis.

Line 305: Based on this estimate of cycle length, the duration of the experiment may

have captured slightly more than 1 cycle. Is this correct? It might be good to state this explicitly and argue that by sampling the cycle on a regular basis, the measurements approximate the mean conditions.

[Reply] L313-316: We appreciate this comment. According to the comment, we have revised the sentence and now explicitly state that our experimental run was certainly longer than one oscillation cycle.